# MPPN: Multi-Resolution Periodic Pattern Network For Long-Term Time Series Forecasting

## Abstract

Long-term time series forecasting plays an important role in various real-world scenarios. Recent deep learning methods for long-term series forecasting tend to capture the intricate patterns of time series by Transformer-based or sampling-based methods. However, most of the extracted patterns are relatively simplistic and may include unpredictable noise. Moreover, the multivariate series forecasting methods usually ignore the individual characteristics of each variate, which may affect the prediction accuracy. To capture the intrinsic patterns of time series, we propose a novel deep learning network architecture, named Multi-resolution Periodic Pattern Network (MPPN), for long-term series forecasting. We first construct context-aware multi-resolution semantic units of time series and employ multi-periodic pattern mining to capture the key patterns of time series. Then, we propose a channel adaptive module to capture the multivariate perceptions towards different patterns. In addition, we adopt an entropy-based method for evaluating the predictability of time series and providing an upper bound on the prediction accuracy before forecasting. Our experimental evaluation on nine real-world benchmarks demonstrated that MPPN significantly outperforms the state-of-the-art Transformer-based, sampling-based and pre-trained methods for long-term series forecasting.

## 1 Introduction

Time series forecasting is a long-standing problem and has been widely used in weather forecasting, energy management, traffic flow scheduling, and financial planning. Long-term time series forecasting (LTSF) means predicting further into the future, which can provide sufficient reference for long-term planning applications and is of great importance. This paper focuses on long-term time series forecasting problem. Most of the typical methods for LTSF task before treated time series as a sequence of values, similar to the sequence in speech and natural language processing. Specifically, the encoding of a lookback window of historical time series values, along with time feature embedding (e.g., Hour of Day, Day of Week and Day of Month) and positional encoding, are combined as the model input sequence. Then the convolution-based Wang et al. (2023) or Transformer-based techniques Zhou et al. (2021); Liu et al. (2021) are used to extract the intricate correlations or high-dimensional features of time series to achieve long-term sequence prediction.

Unlike other types of sequential data, time series data only record scalars at each moment. Data of solitary time points cannot provide adequate semantic information and might contain noise. Therefore, some works implement sub-series Wu et al. (2021) or segments Wang et al. (2023); Zhang & Yan (2023) as the basic semantic tokens aiming to capture the inherent patterns of time series. However, the patterns of time series are intricate and usually entangled and overlapped with each other, which are extremely challenging to clarify. Without making full use of the properties of time series (e.g., period), relying solely on the self-attention or convolution techniques to capture the overlapped time series patterns can hardly avoid extracting noisy patterns. In addition, most of the multivariate time series prediction methods Liu et al. (2022b); Zhang & Yan (2023) mainly focus on modeling the correlations between variates and ignore the individual characteristics of each variate, which may affect the prediction accuracy.

Existing methods for LTSF tasks often involve building complex models based on multi-level time series decomposition or sampling techniques to capture patterns within the time series. Decomposition-based methods attempt to decompose the time series into more predictable parts and predict them separately before aggregating the results Wu et al. (2021); Zhou et al. (2022); Wang et al. (2023); Oreshkin et al. (2019); Zeng et al. (2023). For instance, FEDformer Zhou et al. (2022) and MICN Wang et al. (2023) proposed multi-scale hybrid decomposition approach based on Moving Average to extract various seasonal and trend-cyclical parts of time series. However, the real-world time series are usually intricate which are influenced by multiple factors and can be hardly disentangled. Most sampling-based methods implement downsampling techniques, which can partially degrade the complexity of time series and improve the predictability of the original series Liu et al. (2022a); Zhang et al. (2022). But they can easily suffer from the influence of outliers or noise in time series, which reduces the quality of the extracted patterns and affects their performance in LTSF tasks. Thus, we consider whether it is feasible to extract the characteristics or patterns of a time series explicitly, without relying on decomposition or sampling based approaches.

We believe that, analogous to speech and natural language, time series have their own distinctive patterns that can represent them. The challenge lies in how to extract these patterns. For the same variate, we observe that time series with larger resolutions often exhibit stronger periodicity, whereas those with smaller resolutions tend to have more fluctuations, as shown in Figure 1. Motivated by this, we thought that a time series can be seen as an overlay of multi-resolution patterns. Moreover, time series possess regular patterns, which is why we can predict them. One obvious observation is that real-world time series, such as electricity consumption and traffic, usually exhibit daily and weekly periods. Therefore, we attempt to capture the multi-periodicity of time series to decode their unique characteristics. Further, for multivariate series prediction task, each variate has its own characteristics and perception of temporal patterns. Existing methods frequently employ the same model parameters, which can only model the commonalities among the multiple variates, without taking into account the individualities of each variate.

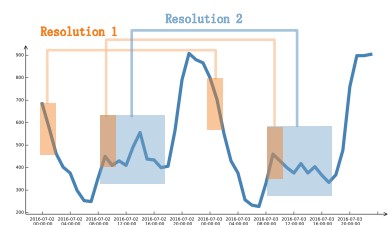

Figure 1: An example of time series with the multi-resolution periodic patterns. It displays a client's electricity consumption for two days and shows that the series generally peaks between 8 PM and midnight, while during daytime, it maintains a low range, presumably implying that the resident is outside working.

Based on the above motivations, we propose a novel deep learning network architecture, named Multi-resolution Periodic Pattern Network (MPPN) for long-term time series forecasting. Firstly, we construct context-aware multi-resolution semantic units of the time series and propose a multi-periodic pattern mining mechanism for capturing the distinctive patterns in time series. Secondly, we propose a channel adaptive module to infer the variate embedding (attributes) from data during training and to perform adaptive weighting on the mined patterns. In addition, we argue that before predicting a time series, it should be evaluated whether the series is predictable or not. Therefore, in this paper, we adopt an entropy-based method for evaluating the predictability of time series and providing an upper bound on how predictable the time series is before carrying out predictions. Our objective is devising an efficient and effective long-term forecasting model, aiming at capturing the intrinsic characteristics of time series. The contributions of this paper are summarized as follows:

- We propose a novel framework MPPN to explicitly capture the inherent multi-resolution and multi-periodic patterns of time series for efficient and accurate long-term series forecasting.

- We propose a channel adaptive module to adaptively model different perceptions of multivariate series towards various temporal patterns, further improving the prediction performance.

- Experimental evaluations on nine real-world benchmarks demonstrate that our MPPN significantly outperforms the state-of-the-art methods in LTSF tasks, while maintaining linear computational complexity. Furthermore, to the best of our knowledge, we are the first to derive predictability results of these widely-used datasets for LTSF tasks.

## 2 RELATED WORK

In the past several decades, numerous methods for time series forecasting have been developed, evolving from conventional statistics (such as ARIMA Williams & Hoel (2003)) and machine learning (such as Prophet Taylor & Letham (2018)) to the current deep learning. Especially, deep learning has gained popularity owing to its strong representation ability and nonlinear modeling capacity. Typical deep learning-based methods include RNN Lai et al. (2018), TCN Bai et al. (2018) and Transformer Vaswani et al. (2017). Transformer-based methods with self-attention mechanism are frequently used for LTSF task Zhou et al. (2022); Wu et al. (2021); Zhang & Yan (2023); Zhou et al. (2021); Liu et al. (2021). Although Transformer-based methods have achieved impressive performance, recent research Zeng et al. (2023) have questioned whether they are suitable for LTSF tasks, especially since the permutation-invariant self-attention mechanism causes loss of temporal information. They have shown that an embarrassingly simple linear model outperforms all Transformer-based models. This highlights the importance of focusing on intrinsic properties of time series.

Recently, sampling-based methods in conjunction with convolution have achieved remarkable results for LTSF tasks. SCINet Liu et al. (2022a) adopts a recursive downsample-convolve-interact architecture that downsamples the sequence into two sub-sequences (odd and even) recursively to extract time series patterns. MICN Wang et al. (2023) implements a multi-scale branch structure with down-sampled convolution for local features extraction and isometric convolution for capturing global correlations. Although these methodologies exhibit better performance compared to Transformer-based models in LTSF task, they neglect intrinsic properties of time series and patterns extracted based on global indiscriminate downsampling may contain noise. With the explosive growth of large models, foundation models have demonstrated excellent performance in NLP and vision fields Devlin et al. (2018); Dosovitskiy et al. (2020); He et al. (2022); Brown et al. (2020). The field of time series analysis has also shifted focus towards developing pre-trained models Zerveas et al. (2021); Nie et al. (2023); Wu et al. (2022), which have shown promising outcomes.

## 3 METHODOLOGY

In this section, we first present the problem definition of the multivariate time series forecasting task and introduce a quantitative evaluation of predictability. Then we introduce our proposed MPPN method. The overall architecture of the MPPN model is illustrated in Figure 2. It consists of Multi-resolution Periodic Pattern Mining (MPPM), a channel adaptive module, and an output layer.

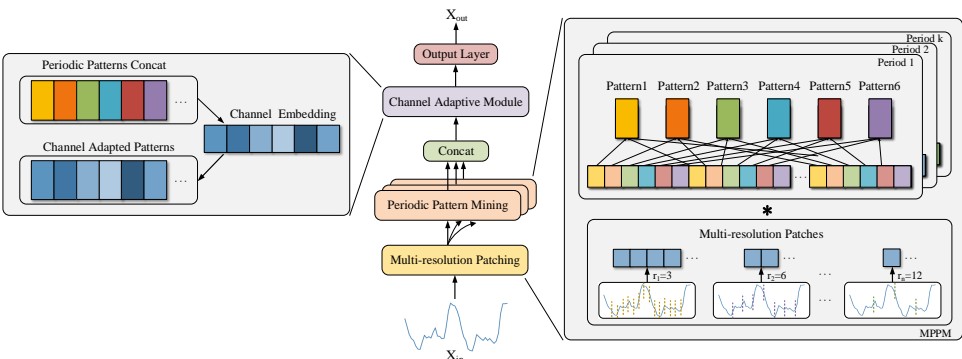

Figure 2: The overall architecture of Multi-resolution Periodic Pattern Network (MPPN).

### 3.1 PROBLEM DEFINITION

Multivariate time series prediction aims to forecast future values of multiple variates based on their historical observations. Considering a multivariate time series $\boldsymbol{X} = [\boldsymbol{x}_1, \ldots, \boldsymbol{x}_t, \ldots, \boldsymbol{x}_T]^T \in \mathbb{R}^{T \times C}$ consisting of $T$ time steps and $C$ recorded variates, where $\boldsymbol{x}_t \in \mathbb{R}^C$ represents an observation of the multivariate time series at time step $t$. We set the look-back window length as $L$ and the length

of the forecast horizon as $H$. Then, given the historical time series $\boldsymbol{X}_{in} = [\boldsymbol{x}_{t-L}, \ldots, \boldsymbol{x}_{t-1}]^T \in \mathbb{R}^{L \times C}$, the forecasting objective is to learn a mapping function $\mathcal{F}$ that predicts the values for the next $H$ time steps $\boldsymbol{X}_{out} = [\boldsymbol{x}_t, \ldots, \boldsymbol{x}_{t+H}]^T \in \mathbb{R}^{H \times C}$:

$$[\boldsymbol{x}_{t-L}, \ldots, \boldsymbol{x}_{t-1}]^T \xrightarrow{\mathcal{F}} [\boldsymbol{x}_t, \ldots, \boldsymbol{x}_{t+H}]^T. \tag{1}$$

## 3.2 PREDICTABILITY

Predictability is a measure that quantifies the confidence in the predictive capability for a time series, providing an upper bound on the accuracy possibly achieved by any forecasting approach. As the foundation of time series prediction, predictability explains to what extent the future can be foreseen, which is often overlooked by prior deep learning-based temporal forecasting methods. In the context of time series prediction, the foremost importance lies not in the construction of predictive models, but rather in the determination of whether the time series itself is predictable. Based on the determinations, it becomes possible to filter out time series with low predictability, such as random walk time series, thereby discerning the meaningfulness of the prediction problem at hand. There exists a multitude of seminal works in the domain of predictability Song et al. (2010); Xu et al. (2019); Guo et al. (2021); Smith et al. (2014), among which the most commonly employed approach is based on entropy measures.

For long-term time series forecasting, we firstly evaluate the predictability following the method in Song et al. (2010), which explored the predictability of human mobility trajectories using entropy rates. Firstly, we discretize the continuous time series into $Q$ discrete values. Denote $\boldsymbol{x} = \{x_1, x_2, \cdots, x_n\}$ as a time series after discretization, its entropy rate is defined as follows:

$$\mathcal{H}_u(\boldsymbol{x}) = \lim_{n \to \infty} \frac{1}{n} \sum_{i=1}^{n} H(x_i \mid x_{i-1}, \cdots, x_1), \tag{2}$$

which characterizes the average conditional entropy $H$ of the current variable given the values of all the past variables as $n \to \infty$. In order to calculate this theoretical value, we utilize an estimator based on Lempel-Ziv encoding Kontoyiannis et al. (1998), which has been proven to be a consistent estimator of the real entropy rate $\mathcal{H}_u(\boldsymbol{x})$. For $\boldsymbol{x}$, the entropy rate $\mathcal{H}_u(\boldsymbol{x})$ is estimated by

$$S = \left(\frac{1}{n} \sum_{i=1}^{n} \Lambda_i\right)^{-1} \ln(n), \tag{3}$$

where $\Lambda_i$ signifies the minimum length of the sub-string starting at position $i$ that has not been encountered before from position $1$ to $i-1$. We further derive the upper bound of predictability $\Pi^{\max}$ by solving the following Fano's inequality Kontoyiannis et al. (1998):

$$S \leq H(\Pi^{\max}) + (1 - \Pi^{\max}) \log_2(Q - 1), \tag{4}$$

where $H(\Pi^{\max}) = -\Pi^{\max} \log_2(\Pi^{\max}) - (1 - \Pi^{\max}) \log_2(1 - \Pi^{\max})$ represents the binary entropy function and $Q$ is the number of distinct values in $\boldsymbol{x}$. It is worth noting that the inequality equation 4 is tight, in the sense that the upper bound of predictability is attainable by some actual algorithm. As a result, the upper bound of predictability provides a theoretical guarantee for conducting long-term time series forecasting.

## 3.3 MULTI-RESOLUTION PERIODIC PATTERN MINING

The MPPM is composed of two key components, namely multi-resolution patching and periodic pattern mining, which are specially designed to capture intricate multi-resolution patterns inherent in time series data with multiple periods. For simplicity, we omit the channel dimension $C$ and denote the hidden state of the series as $D$.

**Multi-resolution patching**    To capture the multi-resolution patterns in time series data, we first obtain context-aware semantic units of the time series. Specifically, as shown in Figure 2, we employ non-overlapping multi-scale convolutional kernels ( inception mechanism Szegedy et al. (2016)) to partition the input historical time series $\boldsymbol{X}_{in}$ into multi-resolution patches. For instance, for a time series with a granularity of 1 hour and assuming a resolution of 3 hours, the above-mentioned

convolution with a kernel size 3 is used to map $\boldsymbol{X}_{in}$ to the output $\boldsymbol{X}_r$. This process can be formulated as follows:

$$\boldsymbol{X}_r = \text{Conv} \, 1d \left(\text{Padding} \left(\boldsymbol{X}_{in}\right)\right)_{\text{kernel} = r}, \tag{5}$$

where $r$ denotes the convolutional kernel size that correspond to the pre-defined temporal resolution. For Conv1d, we set the $kernel$ and $stride$ both to be $r$. For the resolution selection, we choose a set of reasonable resolutions $r \in \{r_1, r_2, \cdots, r_n\}$ based on the granularity of the input time series (See Appendix A.4 for more details). $\boldsymbol{X}_r$ denotes the obtained semantic units of the time series corresponding to resolution $r$.

**Periodic pattern mining**  We implement periodic pattern mining to explicitly capture the multi-resolution and multi-periodic patterns in time series data. We firstly employ Fast Fourier Transform (FFT) to calculate the periodicity of the original time series, following the periodicity computation method proposed by Wu et al. Wu et al. (2022). Briefly, we take the Fourier transform of the original time series $\boldsymbol{X}$ and calculate the amplitude of each frequency. We then select the top-$k$ amplitude values and obtain a set of the most significant frequencies $\{f_1, \cdots, f_k\}$ corresponding to the selected amplitudes, where $k$ is a hyperparameter. Consequently, we acquire $k$ periodic lengths $\{\text{Period}_1, \cdots, \text{Period}_k\}$ that correspond to these frequencies. Similar to Wu et al. (2022), we only consider frequencies within $\left\{1, \cdots, \left[\frac{T}{2}\right]\right\}$. The process is summarized as follows:

$$\mathbf{A} = \text{Avg} \left(\text{Amp} \left(\text{FFT} \left(\boldsymbol{X}\right)\right)\right), \{f_1, \cdots, f_k\} = \underset{f_* \in \left\{1, \cdots, \left[\frac{T}{2}\right]\right\}}{\arg \text{Topk}} \left(\mathbf{A}\right), \text{Period}_i = \left\lceil \frac{T}{f_i} \right\rceil, \tag{6}$$

where $i \in \{1, \cdots, k\}$, FFT$(\cdot)$ represents the FFT, and Amp$(\cdot)$ denotes the amplitude calculation. $\mathbf{A} \in \mathbb{R}^T$ denotes the calculated amplitudes, which are determined by taking the average of $C$ variates using Avg$(\cdot)$.

We then utilize the periodicity calculated above and employ dilated convolutions to achieve multi-periodic and multi-resolution pattern mining. Specifically, given a periodic length of $\text{Period}_i$ and the resolution $r$, we set convolution dilation as $\left\lfloor \frac{\text{Period}_i}{r} \right\rfloor$ and kernel size as $\left\lfloor \frac{L}{\text{Period}_i} \right\rfloor$ for the convolution operation on $\boldsymbol{X}_r$. To obtain regularized patterns, we perform truncation on the outputs of dilated convolution. The process can be formulated as follows:

$$\boldsymbol{X}_{\text{Period}_i, r} = \text{Truncate} \left(\text{Conv} \, 1d \left(\boldsymbol{X}_r\right)_{\text{kernel} = \left\lfloor \frac{L}{\text{Period}_i} \right\rfloor, \text{dilation} = \left\lfloor \frac{\text{Period}_i}{r} \right\rfloor}\right), \tag{7}$$

where $\boldsymbol{X}_{\text{Period}_i, r} \in R^{\left\lfloor \frac{\text{Period}_i}{r} \right\rfloor \times D}$ denotes the patterns extracted corresponding to $\text{Period}_i$ and resolution $r$. For the same period, we concatenate all the corresponding $\boldsymbol{X}_{\text{Period}_i, r}$ of different resolutions $r$ to obtain its whole pattern $\boldsymbol{X}_{\text{Period}_i}$. We then concatenate the patterns of multiple periods to obtain the final multi-periodic pattern of the time series, denoted as $\boldsymbol{X}_{\text{Pattern}} \in R^{P \times D}$, formulated as follows:

$$\boldsymbol{X}_{\text{Period}_i} = \overset{n}{\underset{j=1}{\|}} \boldsymbol{X}_{\text{Period}_i, r_j}, \, \boldsymbol{X}_{\text{Pattern}} = \overset{k}{\underset{i=1}{\|}} \boldsymbol{X}_{\text{Period}_i}, \, P = \sum_{i=1}^{k} \sum_{j=1}^{n} \left\lfloor \frac{\text{Period}_i}{r_j} \right\rfloor. \tag{8}$$

## 3.4 CHANNEL ADAPTIVE MODULE

To achieve adaptivity for each variate, we propose a channel adaptive mechanism. We firstly define a learnable variate embeddings matrix $\boldsymbol{E} \in R^{C \times P}$, which can be updated during model training, where $P$ represents the number of pattern modes extracted by the above MPPM. Next, we apply the sigmoid function to activate the learned variate representation $\boldsymbol{E}$ and then perform broadcasting multiplication with the obtained multi-resolution periodic pattern $\boldsymbol{X}_{\text{Pattern}}$, producing the final channel adaptive patterns $\boldsymbol{X}_{\text{AdpPattern}} \in R^{C \times P \times D}$, formulated as follows:

$$\boldsymbol{X}_{\text{AdpPattern}} = \boldsymbol{X}_{\text{Pattern}} \cdot \text{sigmoid} \left(\boldsymbol{E}\right), \tag{9}$$

At last, we implement the output layer with one fully connected layer to generate the final long-term prediction $\boldsymbol{X}_{out} \in \mathbb{R}^{H \times C}$. The output layer can be formulated as follows:

$$\boldsymbol{X}_{out} = \text{Reshape} \left(\boldsymbol{X}_{\text{AdpPattern}}\right) \cdot \boldsymbol{W} + \boldsymbol{b}, \tag{10}$$

where $\boldsymbol{W} \in \mathbb{R}^{(PD) \times H}$ and $\boldsymbol{b} \in \mathbb{R}^H$ are learnable parameters. $\boldsymbol{X}_{out}$ is the final output of the MPPN. Finally, we adopt the Mean Squared Error (MSE) as the training loss to optimize the model.

## 4 EXPERIMENTS

In this section, we present the experimental evaluation of our MPPN model compared to state-of-the-art baseline models. Additionally, we conduct comprehensive ablation studies and perform model analysis to demonstrate the effectiveness of each module in MPPN. More detailed information can be found in the Appendix.

### 4.1 EXPERIMENTAL SETTINGS

**Datasets**  We conduct extensive experiments on nine widely-used time series datasets, including four *ETT* Zhou et al. (2021) (ETTh1, ETTh2, ETTm1, ETTm2), *Electricity*, *Exchange-Rate* Lai et al. (2018), *Traffic*, *Weather* and *ILI* dataset. A brief description of these datasets is presented in Table 1. We provide a detailed dataset description in Appendix A.1.

Table 1: Dataset statistics.

| Datasets | Electricity | Weather | Traffic | Exchange-Rate | ILI | ETTh1&ETTh2 | ETTm1&ETTm2 |
|---|---|---|---|---|---|---|---|
| Timesteps | 26,304 | 52,696 | 17,544 | 7,588 | 966 | 17,420 | 69,680 |
| Features | 321 | 21 | 862 | 8 | 7 | 7 | 7 |
| Granularity | 1hour | 10min | 1hour | 1day | 1week | 1hour | 15min |

**Baselines**  We employ two pre-trained models: PatchTST Nie et al. (2023) and TimesNet Wu et al. (2022), two SOTA Linear-based models: DLinear and NLinear Zeng et al. (2023), three cutting-edge Transformer-based models: Crossformer Zhang & Yan (2023), FEDformer Zhou et al. (2022), Autoformer Wu et al. (2021), and two CNN-based models: MICN Wang et al. (2023) and SCINet Liu et al. (2022a) as baselines. We choose the PatchTST/64 due to its superior performance compared to PatchTST-42. For FEDformer, we select the better one (FEDformer-f, which utilizes Fourier transform) for comparison. More information about the baselines can be found in Appendix A.3.

**Implementation details**  Our model is trained with L2 loss, using the ADAM Kingma & Ba (2014) optimizer with an initial learning rate of 1e-3 and weight decay of 1e-5. The training process is early stopped if there is no loss reduction on the validation set after three epochs. All experiments are conducted using PyTorch and run on a single NVIDIA Tesla V100 GPU. Following previous work Zhou et al. (2022); Wu et al. (2021); Wang et al. (2023), we use Mean Square Error (MSE) and Mean Absolute Error (MAE) as evaluation metrics. See Appendix A.4 for more detailed information.

### 4.2 MAIN RESULTS

Table 2: Predictability and periodicity results of the nine benchmark datasets.

| Datasets | Electricity | Weather | Traffic | Exchange-Rate | ILI | ETTh1 | ETTh2 | ETTm1 | ETTm2 |
|---|---|---|---|---|---|---|---|---|---|
| Timesteps | 26,304 | 52,696 | 17,544 | 7,588 | 966 | 17,420 | 17,420 | 69,680 | 69,680 |
| Predictability | 0.876 | 0.972 | 0.934 | 0.973 | 0.917 | 0.853 | 0.927 | 0.926 | 0.967 |
| Top-1 period | 24 | 144 | 12 | – | – | 24 | – | 96 | – |

**Predictability analysis**  As a prerequisite, we investigate the predictability of the nine public datasets before constructing prediction models. For each dataset, a quantitative metric is provided in accordance with the method outlined in Section 3.2. We designate the average predictability of distinct univariate datasets as the measure of predictability for each benchmark dataset in question. The corresponding results are summarized in Table 2. It can be observed from Table 2 that all predictability results exceed 0.85, indicating the nine benchmark datasets exhibit a notable level of predictability. This provides sufficient confidence and theoretical assurance for constructing excellent prediction models upon the nine benchmark datasets. Results of MPPN in Table 3 show that MAE and MSE for Weather and Exchange with predictability larger than 0.97 reside at a low level, while those for ETTm1 and ILI with lower predictability lie at a relatively higher level. Although the situation is not always the case, the general rule is that for datasets with higher predictability,

carefully constructed predictive models usually tend to exhibit lower prediction metrics (e.g., MSE or MAE).

**Periodicity analysis** To capture the intricate temporal patterns and multiple-periodicity within the time series, we employ the method of FFT as discussed in Section 3 to extract the Top-1 period for each of the nine benchmark datasets. See the last row of Table 2 for details. It can be concluded from Table 2 that Electricity, Weather, Traffic, ILI, ETTh1, and ETTm1 exhibit noticeable periodic patterns, while the remaining datasets do not possess discernible periodicities, indicating the manifestation of more complex time series patterns.

Table 3: **Multivariate** long-term time series forecasting results with different prediction length $O \in \{24, 36, 48, 60\}$ for ILI dataset and $O \in \{96, 192, 336, 720\}$ for others. The SOTA results are **bolded**, while the sub-optimal results are underlined.

| Models | | MPPN | | PatchTST/64 | | NLinear | | DLinear | | TimesNet | | SCINet | | MICN | | FEDformer | | Autoformer | | Crossformer | |
|---|---|---|---|---|---|---|---|---|---|---|---|---|---|---|---|---|---|---|---|---|---|
| Metric | | MSE | MAE | MSE | MAE | MSE | MAE | MSE | MAE | MSE | MAE | MSE | MAE | MSE | MAE | MSE | MAE | MSE | MAE | MSE | MAE |
| Weather | 96 | **0.144** | **0.196** | 0.149 | 0.198 | 0.182 | 0.232 | 0.174 | 0.233 | 0.170 | 0.220 | 0.184 | 0.242 | 0.170 | 0.235 | 0.219 | 0.300 | 0.263 | 0.332 | 2.320 | 0.877 |
| | 192 | **0.189** | **0.240** | 0.194 | 0.241 | 0.225 | 0.269 | 0.218 | 0.278 | 0.224 | 0.263 | 0.244 | 0.298 | 0.223 | 0.285 | 0.271 | 0.331 | 0.295 | 0.354 | 2.370 | 0.919 |
| | 336 | **0.240** | **0.281** | 0.245 | 0.282 | 0.271 | 0.301 | 0.263 | 0.314 | 0.282 | 0.304 | 0.287 | 0.322 | 0.278 | 0.339 | 0.318 | 0.354 | 0.346 | 0.385 | 3.839 | 1.243 |
| | 720 | **0.310** | **0.333** | 0.314 | 0.334 | 0.339 | 0.349 | 0.332 | 0.374 | 0.357 | 0.353 | 0.346 | 0.360 | 0.342 | 0.386 | 0.410 | 0.419 | 0.428 | 0.433 | 5.161 | 1.494 |
| Traffic | 96 | **0.387** | **0.271** | 0.393 | 0.284 | 0.412 | 0.282 | 0.413 | 0.287 | 0.589 | 0.351 | 0.444 | 0.281 | 0.524 | 0.307 | 0.588 | 0.368 | 0.644 | 0.415 | 0.530 | 0.293 |
| | 192 | **0.396** | **0.273** | 0.408 | 0.290 | 0.425 | 0.287 | 0.424 | 0.290 | 0.618 | 0.324 | 0.528 | 0.321 | 0.541 | 0.315 | 0.606 | 0.373 | 0.623 | 0.386 | 0.560 | 0.307 |
| | 336 | **0.410** | **0.279** | 0.419 | 0.294 | 0.437 | 0.293 | 0.438 | 0.299 | 0.635 | 0.341 | 0.531 | 0.321 | 0.540 | 0.312 | 0.629 | 0.390 | 0.620 | 0.385 | 0.535 | 0.324 |
| | 720 | **0.449** | **0.301** | 0.508 | 0.366 | 0.465 | 0.311 | 0.466 | 0.316 | 0.664 | 0.352 | 0.620* | 0.394* | 0.599 | 0.324 | 0.627 | 0.381 | 0.677 | 0.418 | 0.591 | 0.314 |
| Electricity | 96 | 0.131 | **0.226** | **0.129** | **0.222** | 0.141 | 0.237 | 0.140 | 0.237 | 0.168 | 0.271 | 0.167 | 0.269 | 0.163 | 0.269 | 0.189 | 0.305 | 0.202 | 0.317 | 0.220 | 0.303 |
| | 192 | **0.145** | **0.239** | 0.149 | 0.244 | 0.154 | 0.249 | 0.154 | 0.250 | 0.191 | 0.293 | 0.174 | 0.280 | 0.178 | 0.286 | 0.197 | 0.312 | 0.233 | 0.338 | 0.279 | 0.347 |
| | 336 | **0.162** | **0.256** | 0.163 | 0.259 | 0.171 | 0.265 | 0.169 | 0.268 | 0.197 | 0.299 | 0.186 | 0.292 | 0.184 | 0.293 | 0.214 | 0.329 | 0.260 | 0.359 | 0.336 | 0.370 |
| | 720 | **0.200** | **0.289** | 0.199 | 0.292 | 0.210 | 0.298 | 0.204 | 0.300 | 0.262 | 0.340 | 0.231* | 0.316* | 0.214 | 0.324 | 0.244 | 0.351 | 0.256 | 0.361 | 0.429 | 0.441 |
| Exchange | 96 | 0.089 | **0.204** | 0.099 | 0.223 | 0.089 | 0.208 | 0.085 | 0.209 | 0.105 | 0.235 | 0.116 | 0.254 | **0.082** | 0.205 | 0.134 | 0.262 | 0.143 | 0.273 | 0.257 | 0.383 |
| | 192 | 0.177 | 0.295 | 0.226 | 0.343 | 0.180 | 0.300 | 0.162 | 0.296 | 0.234 | 0.352 | 0.218 | 0.345 | **0.157** | 0.298 | 0.261 | 0.372 | 0.271 | 0.380 | 0.878 | 0.732 |
| | 336 | 0.344 | 0.418 | 0.348 | 0.430 | 0.331 | 0.415 | 0.333 | 0.441 | 0.387 | 0.457 | 0.294 | 0.413 | **0.269** | **0.402** | 0.442 | 0.494 | 0.456 | 0.506 | 1.149 | 0.858 |
| | 720 | 0.929 | 0.731 | 1.225 | 0.834 | 0.943 | 0.728 | 0.898 | 0.725 | 0.936 | 0.737 | 1.110 | 0.767 | **0.701** | **0.653** | 1.125 | 0.820 | 1.090 | 0.812 | 1.538 | 1.002 |
| ILI | 24 | 1.796 | **0.860** | **1.794** | 0.913 | 2.285 | 0.983 | 2.280 | 1.061 | 2.110 | 0.948 | 3.409 | 1.245 | 3.031 | 1.180 | 3.221 | 1.242 | 3.410 | 1.296 | 3.197 | 1.199 |
| | 36 | 1.748 | **0.840** | **1.478** | **0.840** | 2.119 | 0.938 | 2.235 | 1.059 | 2.764 | 1.035 | 3.200 | 1.204 | 2.507 | 1.011 | 2.660 | 1.071 | 3.365 | 1.252 | 3.191 | 1.210 |
| | 48 | **1.692** | **0.840** | 1.849 | 0.917 | 2.062 | 0.933 | 2.298 | 1.079 | 2.461 | 0.963 | 2.943 | 1.187 | 2.427 | 1.013 | 2.717 | 1.101 | 3.125 | 1.200 | 3.450 | 1.205 |
| | 60 | **1.840** | **0.881** | 1.971 | 0.954 | 2.258 | 0.994 | 2.573 | 1.157 | 2.218 | 0.931 | 2.719 | 1.189 | 2.654 | 1.085 | 2.840 | 1.148 | 2.847 | 1.146 | 3.505 | 1.264 |
| ETTh1 | 96 | **0.371** | **0.393** | 0.372 | 0.403 | 0.374 | 0.394 | 0.384 | 0.405 | 0.389 | 0.412 | 0.405 | 0.428 | 0.405 | 0.431 | 0.377 | 0.416 | 0.436 | 0.448 | 0.426 | 0.436 |
| | 192 | **0.405** | **0.413** | 0.413 | 0.429 | 0.408 | 0.415 | 0.443 | 0.450 | 0.440 | 0.442 | 0.470 | 0.470 | 0.501 | 0.489 | 0.424 | 0.446 | 0.444 | 0.451 | 0.585 | 0.547 |
| | 336 | **0.426** | **0.425** | 0.422 | 0.440 | 0.429 | 0.427 | 0.447 | 0.448 | 0.495 | 0.471 | 0.530 | 0.514 | 0.541 | 0.528 | 0.450 | 0.463 | 0.516 | 0.493 | 0.552 | 0.521 |
| | 720 | **0.436** | **0.452** | 0.447 | 0.468 | 0.440 | 0.453 | 0.504 | 0.515 | 0.518 | 0.495 | 0.584 | 0.561 | 0.822 | 0.700 | 0.474 | 0.491 | 0.500 | 0.501 | 0.655 | 0.604 |
| ETTh2 | 96 | 0.278 | **0.335** | **0.273** | 0.337 | 0.277 | 0.338 | 0.290 | 0.353 | 0.332 | 0.370 | 0.397 | 0.434 | 0.292 | 0.355 | 0.339 | 0.381 | 0.397 | 0.430 | 0.843 | 0.669 |
| | 192 | 0.344 | **0.380** | **0.340** | 0.381 | 0.344 | 0.381 | 0.388 | 0.422 | 0.397 | 0.410 | 0.594 | 0.548 | 0.441 | 0.454 | 0.429 | 0.437 | 0.440 | 0.441 | 0.472 | 0.492 |
| | 336 | 0.362 | 0.400 | **0.329** | **0.384** | 0.357 | 0.400 | 0.463 | 0.473 | 0.453 | 0.451 | 0.615 | 0.559 | 0.545 | 0.515 | 0.445 | 0.461 | 0.477 | 0.481 | 0.898 | 0.687 |
| | 720 | 0.393 | 0.434 | **0.380** | **0.423** | 0.394 | 0.436 | 0.733 | 0.606 | 0.438 | 0.450 | 1.079 | 0.764 | 0.834 | 0.688 | 0.455 | 0.475 | 0.482 | 0.489 | 1.250 | 0.830 |
| ETTm1 | 96 | **0.287** | **0.335** | 0.290 | 0.344 | 0.306 | 0.348 | 0.301 | 0.345 | 0.335 | 0.377 | 0.339 | 0.386 | 0.315 | 0.365 | 0.349 | 0.401 | 0.520 | 0.487 | 0.392 | 0.425 |
| | 192 | **0.330** | **0.360** | 0.334 | 0.371 | 0.349 | 0.375 | 0.336 | 0.366 | 0.405 | 0.411 | 0.381 | 0.413 | 0.361 | 0.388 | 0.390 | 0.423 | 0.543 | 0.498 | 0.472 | 0.492 |
| | 336 | **0.369** | **0.382** | **0.369** | 0.392 | 0.375 | 0.388 | 0.372 | 0.389 | 0.414 | 0.422 | 0.414 | 0.436 | 0.387 | 0.416 | 0.433 | 0.450 | 0.652 | 0.543 | 0.527 | 0.525 |
| | 720 | 0.426 | **0.414** | 0.416 | 0.420 | 0.433 | 0.422 | 0.427 | 0.423 | 0.479 | 0.459 | 0.475 | 0.470 | 0.445 | 0.454 | 0.480 | 0.474 | 0.707 | 0.570 | 0.608 | 0.564 |
| ETTm2 | 96 | **0.162** | **0.250** | 0.166 | 0.256 | 0.167 | 0.255 | 0.172 | 0.267 | 0.188 | 0.266 | 0.196 | 0.294 | 0.178 | 0.272 | 0.189 | 0.280 | 0.254 | 0.321 | 0.360 | 0.426 |
| | 192 | **0.217** | **0.288** | 0.223 | 0.296 | 0.221 | 0.293 | 0.237 | 0.314 | 0.263 | 0.311 | 0.369 | 0.424 | 0.236 | 0.310 | 0.255 | 0.322 | 0.273 | 0.331 | 0.580 | 0.568 |
| | 336 | **0.273** | **0.325** | 0.274 | 0.329 | 0.275 | 0.327 | 0.295 | 0.359 | 0.322 | 0.349 | 0.410 | 0.447 | 0.299 | 0.351 | 0.323 | 0.363 | 0.340 | 0.371 | 1.623 | 0.799 |
| | 720 | 0.368 | **0.383** | **0.362** | 0.385 | 0.370 | 0.385 | 0.427 | 0.439 | 0.424 | 0.408 | 0.583 | 0.535 | 0.435 | 0.452 | 0.421 | 0.419 | 0.453 | 0.439 | 1.954 | 1.015 |

Results* are from SCINet Liu et al. (2022a) due to out-of-memory. Other results are implemented by us.

**Multivariate results** For multivariate long-term forecasting, our proposed MPPN outperforms all baseline models and achieves the state-of-the-art performance on most of the datasets (Table 3). Compared to the up-to-date Transformer-based models, MPPN achieves substantial reductions of **22.43%** in MSE and **17.76%** in MAE. Additionally, when compared to the best results achieved by CNN-based models, MPPN achieves a notable overall reduction of **19.41%** on MSE and **14.77%** on MAE. Regarding the Linear-based models, MPPN consistently outperforms it across all datasets, with particularly notable improvements observed in large datasets(Weather, Traffic) and Exchange-Rate. Specifically, MPPN reduces MSE by **11.48%** in Weather, **5.57%** in Traffic, and **18.80%** in Exchange-Rate. Compared with the SOTA pre-trained models for LTSF tasks, MPPN can still outperform them in general. The effectiveness of pre-trained models stems from their extensive parameters while our model is a lightweight model aiming to capture the intrinsic temporal patterns.

For datasets with evident periodicity, such as Electricity, Weather, and Traffic (Table 2), our MPPN shows stronger capabilities in capturing and modeling their inherent time series patterns compared to other models. As for data without clear periodic patterns, like the Exchange-Rate and ETTh2, MPPN still provides commendable predictions. The impressive pattern mining aptitude of MPPN for LTSF tasks makes it a practical choice for real-world applications. Besides, the overall sampling-based and decomposition-based methods perform better than other baseline models, highlighting the importance of capturing specific patterns in time series data. We also list the full benchmark and univariate long-term forecasting results in Appendix B.

## 4.3 ABLATION STUDIES

Table 4: Ablation studies: multivariate long-term series prediction results on Weather and Electricity with input length 720 and prediction length in $\{96, 192, 336, 720\}$. Three variants of MPPN are evaluated, with the best results highlighted in bold.

| Methods | | MPPN | | w/o multi-resolution | | w/o periodic sampling | | w/o channel adaption | |
|---|---|---|---|---|---|---|---|---|---|
| Metric | | MSE | MAE | MSE | MAE | MSE | MAE | MSE | MAE |
| Weather | 96 | **0.144** | **0.196** | 0.165 | 0.226 | 0.147 | 0.200 | 0.167 | 0.222 |
| | 192 | **0.189** | **0.240** | 0.209 | 0.261 | 0.196 | 0.249 | 0.212 | 0.259 |
| | 336 | **0.240** | **0.281** | 0.258 | 0.302 | 0.246 | 0.289 | 0.258 | 0.295 |
| | 720 | **0.310** | **0.333** | 0.313 | 0.336 | 0.312 | 0.335 | 0.322 | 0.341 |
| Electricity | 96 | **0.131** | **0.226** | 0.156 | 0.264 | 0.133 | 0.228 | 0.133 | 0.228 |
| | 192 | **0.145** | **0.239** | 0.171 | 0.276 | 0.148 | 0.242 | 0.147 | 0.241 |
| | 336 | **0.162** | **0.256** | 0.186 | 0.290 | 0.164 | 0.258 | 0.164 | 0.258 |
| | 720 | **0.200** | **0.289** | 0.223 | 0.319 | 0.203 | 0.292 | 0.203 | 0.291 |

In this section, we conduct ablation studies on Weather and Electricity to assess the effect of each module in MPPN. Three variants of MPPN are evaluated: 1) **w/o multi-resolution**: we remove the multi-resolution patching and instead employ a single resolution and a single period for sampling; 2) **w/o periodic sampling**: we eliminate periodic pattern mining and directly adopt multi-resolution patching followed by a channel adaptive module and an output layer; 3) **w/o channel adaption**: we drop channel adaptive module and treat each channel equally; The experimental results are summarized in Table 4 with best results bolded. As can be seen from Table 4, omitting multi-resolution or periodic pattern mining leads to significant performance degradation. Employing multi-resolution patching and multiple periodic pattern mining facilitates better exploration of the intrinsic patterns in times series. Channel adaption also brings noticeable performance improvement for both datasets. Compared to Electricity containing only electricity consumption data, the impact of channel adaption is more pronounced on Weather. Since Weather dataset comprises distinct meteorological indicators, such as wind velocity and air temperature, it is conducive to regard different channels distinctively rather than treating them equally. Overall, MPPN enjoys the best performance across different datasets and prediction horizons, which demonstrates the effectiveness of each modeling mechanism. Further ablation experiment results can be found in the Appendix.

## 4.4 MODEL ANALYSIS

**Periodic pattern** As shown in Figure 3(a), we randomly select a variate from the Electricity dataset with hourly interval and sample its historical data over 7 days. We find that the data at the same time point for each day exhibits fluctuations within a relatively small range, while the magnitude of the fluctuations varies at different time points. Our findings confirm the existence of periodic patterns in the analysed time series, demonstrating that our proposed MPPM in Section 3 which can extract these patterns could improve the performance. Meanwhile, we also investigate the patterns of three-hour resolution by taking the mean value of the adjacent three time points, as shown in Figure 3(b). Time series data exhibits periodic patterns across different resolutions, thus integrating multi-resolution patterns of the series can enhance modeling accuracy.

**Channel adaptive modeling** To illustrate the effect of the channel adaptive module, we visualize the channel embedding matrix on ETTh1 dataset with eight patterns. We set the look-back window $L = 336$ and the prediction horizon to be 96. In Figure 4, the varying hues and numbers in each block represent the sensitivity of various channels to distinct temporal patterns. It can be seen that most variates (channels) are significantly influenced by the third and fourth patterns, with the exception of 'LULF', which denotes **L**ow **U**se**F**ul **L**oad. The channel adaptive module in MPPN helps capture the perceptions of multivariate towards different patterns, while also providing interpretability to our approach.

**Efficiency analysis** We compare the training time for one epoch of our MPPN with serval baseline models on the Weather dataset, and the results are shown in Figure 5. In general, pre-trained

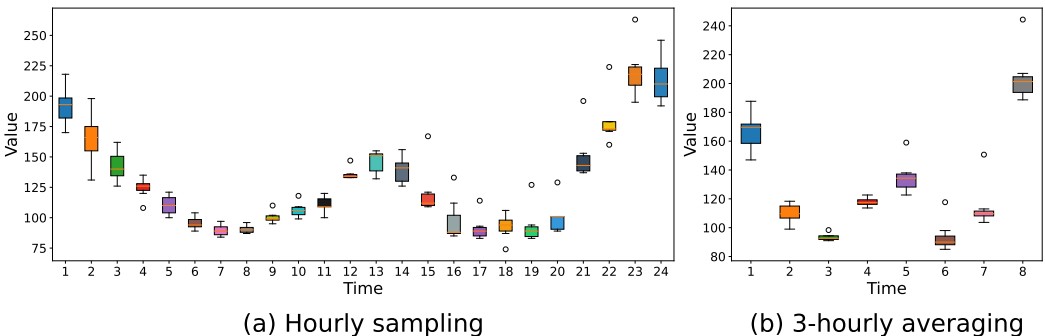

Figure 3: Period pattern analysis on the Electricity dataset.

models exhibit the highest time complexity, followed by Transformer-based models. While MPPN demonstrates sightly higher time complexity compared to the single-layer DLinear, the difference is not significant under the premise of better prediction accuracy. As the prediction length increases, the training time of certain models, such as TimesNet, MICN, and FEDformer, shows a noticeable growth. Meanwhile, models like SCINet and Crossformer do not show a significant increase as the prediction length grows, but they still have considerably higher training time compared to MPPN. Our MPPN model exhibits superior efficiency in handling long-term time series forecasting.

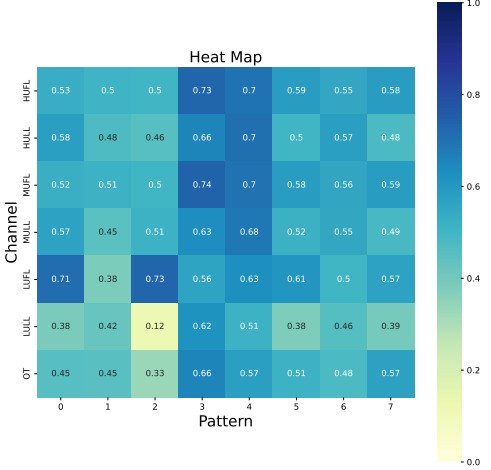

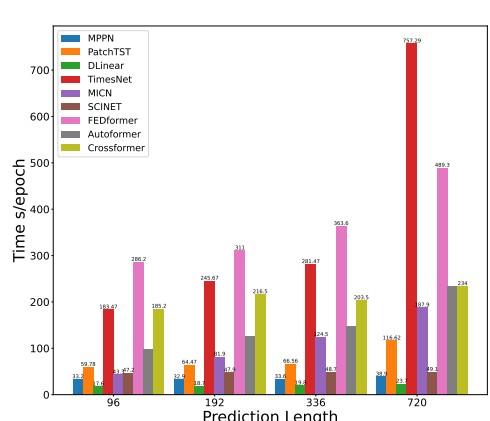

Figure 4: Heat map of channel adaption on ETTh1 with eight extracted patterns.

Figure 5: Comparison of the training time for different baseline models and our MPPN.

## 5 CONCLUSION

In this paper, we propose a novel deep learning network architecture MPPN for long-term time series forecasting. We construct multi-resolution contextual-aware semantic units of time series and propose the multi-period pattern mining mechanism to explicitly capture key time series patterns. Furthermore, we propose a channel-adaptive module to model each variate's perception of different extracted patterns for multivariate series prediction. Additionally, we employ an entropy-based method for evaluating the predictability and providing an upper bound on the prediction accuracy before carrying out predictions. Extensive experiments on nine real-world datasets demonstrate the superiority of our method in long-term forecasting tasks compared to state-of-the-art methods.

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
