# OpenReview forum: "MPPN: Multi-Resolution Periodic Pattern Network  For Long-Term Time Series Forecasting"
_ICLR.cc/2024/Conference — ICLR 2024 Conference Withdrawn Submission_

### Official Review · Reviewer_pRbf · 2023-10-19

**Soundness:** 2 fair
**Presentation:** 1 poor
**Contribution:** 2 fair
**Rating:** 5
**Confidence:** 4

**Summary:**

This paper introduces a novel deep-learning network architecture MPPN for long-term time series forecasting (LTSF), which considers multi-resolution periodic pattern in the time series, and evaluates the model in multiple real-world datasets.

**Strengths:**

1.The overall idea of the paper seems sound.

2.The paper contains a substantial number of experiments.

3.The paper shows figures on the actual prediction produced by different models.

**Weaknesses:**

1.I have reservations regarding the novelty of the paper, what is the main differences between this work and MICN [1]?

2.The paper doesn't compare to recent SOTA models, such as PatchTST [2] and TimesNet [3].

3.According to this paper [4], the Exchange dataset used in the experiment has been deemed invalid as predicting exchange rates over a period of nearly 2 years (720 days) is practically impossible. Hence, beating a naive random walk forecast should be infeasible.

4.Some figures in the paper require improvement. For instance, the texts on the coordinate axis in Figure 1 are difficult to read, and the numbers above the bars in Figure 5 are hard to discern.

5.The performance of MPPN does not seem superior to other models based on the prediction showcases in Figures 6-13.

[1]Wang, Huiqiang, et al. "Micn: Multi-scale local and global context modeling for long-term series forecasting." The Eleventh International Conference on Learning Representations. 2022.

[2]Nie, Yuqi, et al. "A time series is worth 64 words: Long-term forecasting with transformers." arXiv preprint arXiv:2211.14730 (2022).

[3]Wu, H., Hu, T., Liu, Y., Zhou, H., Wang, J., and Long, M. Timesnet: Temporal 2d-variation modeling for general time series analysis. In The Eleventh International Conference on Learning Representations, 2023.

[4]Hewamalage, H., Ackermann, K. & Bergmeir, C. Forecast evaluation for data scientists: common pitfalls and best practices. Data Min Knowl Disc 37, 788–832 (2023).

**Questions:**

Please refer to the Weaknesses.

---

> ### Author Response · Authors · 2023-11-22
> **Response to Reviewer pRbf**
>
> **W1:** I have reservations regarding the novelty of the paper, what is the main differences between this work and MICN [1]?
>
> **A1:** Firstly, MICN utilizes multi-scale hybrid decomposition to decompose the time series into seasonal and trend components. These components are separately predicted and then combined to obtain the final prediction. In contrast, MPPN does not employ such a decomposition.
>
> Secondly, MICN incorporates three types of embeddings for its input representation, including temporal embedding, position embedding, and value embedding. On the other hand, MPPN only utilizes value embedding for the input representation.
>
> Thirdly, MICN uses the proposed multi-scale isometric convolution to capture local and global features, while MPPN does not distinguish between local and global features, and does not use an isometric convolution  (a variant of casual convolution, the convolution kernel size is equal to the length of the input sequence).
>
> Fourthly, MICN applies downsampling (with a sampling interval of {$\{\frac{I}{4}, \frac{I}{8}, ...\}$}, where $I$ represents the input sequence length) followed by upsampling. In contrast, MPPN only performs periodic sampling and does not utilize upsampling. Additionally, the sampling interval for MPPN corresponds to the length of the period.
>
> In addition, as mentioned in the overall response, for the Traffic dataset, MICN has a parameter count that is 100 times larger than MPPN. However, the average prediction Mean Squared Error (MSE)  of MICN is still 30% higher than MPPN. Moreover, as the prediction length increases, the training time of MICN significantly increases, while the training time of MPPN remains almost constant. Please refer to Figure 5 in the paper for further details.
>
> **W2:** The paper doesn't compare to recent SOTA models, such as PatchTST [2] and TimesNet [3].
>
> **A2:** **We have previously conducted a detailed comparison between PatchTST and TimesNet, as outlined in Table 3.** We kindly request you to revisit the information presented in that table.
>
> **W3:** According to this paper [4], the Exchange dataset used in the experiment has been deemed invalid as predicting exchange rates over a period of nearly 2 years (720 days) is practically impossible. Hence, beating a naive random walk forecast should be infeasible.
>
> **A3:** While it is true that predicting exchange rates over an extended period, such as 720 days, poses a significant challenge, it is important to note that the Exchange dataset has been widely employed in the field of time series forecasting. Numerous studies have demonstrated the preddictability of this dataset, showcasing that the development of appropriate prediction methods can yield favorable results even in scenarios requiring forecasts for a prolonged period. See, for example, [1], [2] and [3] for further discussions.
>
> **W4:** Some figures in the paper require improvement. For instance, the texts on the coordinate axis in Figure 1 are difficult to read, and the numbers above the bars in Figure 5 are hard to discern.
>
> **A4:** We will update the images in our paper to enhance the legibility of the accompanying text.
>
> **W5:** The performance of MPPN does not seem superior to other models based on the prediction showcases in Figures 6-13.
>
> **A5:** As illustrated in Figures 6-13, our model adeptly captures both the trend and fluctuations in time series across various temporal resolutions (96, 192, 336, 720) and datasets (ETTh1 and Weather). In contrast, other methods exhibit relatively good performance only under specific settings. For instance, DLinear excels in forecasting ETTh1 under the input-336-predict-96 setting, but its efficacy diminishes notably with an extended forecast length. Similarly, FEDformer demonstrates commendable performance in predicting ETTh1 yet exhibits a markedly higher error when applied to the Weather dataset. In summary, our method consistently outperforms alternative approaches across all datasets and settings.
>
> [1]Wang, Huiqiang, et al. "Micn: Multi-scale local and global context modeling for long-term series forecasting." The Eleventh International Conference on Learning Representations. 2022.
>
> [2]Nie, Yuqi, et al. "A time series is worth 64 words: Long-term forecasting with transformers." arXiv preprint arXiv:2211.14730 (2022).
>
> [3]Wu, H., Hu, T., Liu, Y., Zhou, H., Wang, J., and Long, M. Timesnet: Temporal 2d-variation modeling for general time series analysis. In The Eleventh International Conference on Learning Representations, 2023.
>
> [4]Hewamalage, H., Ackermann, K. & Bergmeir, C. Forecast evaluation for data scientists: common pitfalls and best practices. Data Min Knowl Disc 37, 788–832 (2023).

---

> > ### Comment · Reviewer_pRbf · 2023-11-23
> > **Thank you for your response**
> >
> > Thank you for your response, and you have addressed my second concern (which I hadn't found in prediction showcases before and mistakenly assumed it wasn't compared), and you have provided a detailed explanation of the distinctions from the MICN. However, I find the innovation of the paper still somewhat lacking for ICLR standards, and improvement are needed in both the graphical representations and writing. I have adjusted the score to 5.

---

### Official Review · Reviewer_saoQ · 2023-10-24

**Soundness:** 2 fair
**Presentation:** 3 good
**Contribution:** 2 fair
**Rating:** 3
**Confidence:** 4

**Summary:**

The authors point out that time series can take on multiple-period properties in multiple resolutions. Based on this observation, they propose to Multi-resolution Periodic Pattern Network (MPPN). Using 1d convolutional neural networks with different kernel and dilation sizes, it tries to capture multiple periodicities in multiple resolutions. Furthermore, to recognize different temporal dynamics in different variables, they utilize a channel adaptive module. Meanwhile, they introduce entropy-based methods to analyze the predictability of time series.

**Strengths:**

**A measure for evaluating predictability** is introduced in this paper. This measure helps in determining the suitability of a given dataset for time series forecasting tasks, i.e., assessing whether it is feasible to predict future outcomes based on past observations within the dataset. In instances where time series data contains a significant amount of noise, there is a possibility that it is impossible to find a regular pattern in this time series. Thus, in the realm of time series forecasting, it becomes crucial to distinguish between datasets with clear patterns and those that are highly erratic. The method presented in this paper offers a means to differentiate between well-structured time series datasets and those characterized by substantial noise.

**Weaknesses:**

**Insufficient novelty and contributions**
1. I think the proposed method is just the concatenation of existing several works. In detail, as for multiple resolutions, [2] addresses this problem in a similar way, using 1d CNN with different kernel sizes. Also, the method to find multiple periods in a time series is the same as that of [1]. Finally, a channel adaptive module is almost similar to [3]. Can you give more explanations that your method is not just the concatenation of existing ones?

**More explanations**
1. Why do sampling-based methods easily suffer from the influences of noise? Also, I'm curious about why these kinds of methods neglect the intrinsic properties of time series. Although periodicity is not modeled explicitly, it can still be considered.

2. The authors argue that "Without making full use of the properties of time series (e.g., period), relying solely on the self-attention or convolution techniques to capture the overlapped time series patterns can hardly avoid extracting noisy patterns". I think the use of the periodic property is not directly connected to reducing noise. Some papers have to be cited to make connections.

3. Why do results in [6]  highlight the importance of focusing on the intrinsic properties of time series?

4. Can you give the reason why you set kernel and dilation size to $\frac{L}{Period_{i}}$ and $\frac{Period_{i}}{r}$?

5. In Table 2, the authors provide predictability of each dataset based on Section 3.2. Can you give a more detailed formula for predictability, such as how to identify sub-strings encountered before?

[1] Wu et al., TimesNet: Temporal 2D-Variation Modeling for General Time Series Analysis, 2023, ICLR
[2] Liu et al., Pyraformer: Low-Complexity Pyramidal Attention for Long-Range Time Series Modeling and Forecasting, 2022, ICLR
[3] Shao et al., Spatial-Temporal Identity: A Simple yet Effective Baseline for Multivariate Time Series Forecasting, 2022, CIKM
[4] Wang et al., MICN: MULTI-SCALE LOCAL AND GLOBAL CONTEXT MODELING FOR LONG-TERM SERIES FORECASTING, 2023, ICLR
[5] Zhang et al., Crossformer: Transformer Utilizing Cross-Dimension Dependency for Multivariate Time Series Forecasting, 2023, ICLR
[6] Zeng et al., Are Transformers Effective for Time Series Forecasting?, 2023, AAAI

**Questions:**

Refer to the 'Weakness' section

---

> ### Author Response · Authors · 2023-11-22
> **Response to Reviewer saoQ**
>
> **W1:** I think the proposed method is just the concatenation of existing several works. In detail, ... your method is not just the concatenation of existing ones?
>
> **A1:** Firstly, none of these methods focus on extracting time series patterns explicitly. The existence of periodicity and multi-resolution is a fundamental characteristic of time series, so it is not surprising that previous works, such as TimesNet [1] and [2], have utilized these properties. We have thoroughly analyzed TimesNet in the overall response. As for [2], they construct a multi-resolution C-ary tree, using the Coarser-Scale Construction Module (CSCM), where nodes at a coarser scale encapsulate the information derived from C nodes at the corresponding finer scale. We opted against employing a tree structure and instead chose a more flexible resolution based on the input time series. Utilizing one-dimensional convolution, we model the multi-resolution patterns of the time series data.
>
> In [3], they incorporate concatenated spatial embedding, temporal embedding, and value embedding to directly map the output time series prediction using multilayer perceptrons and regression layers. This approach differs significantly from the concept of our Channel Adaptive Module. Furthermore, [3] only conducts short-term time series prediction experiments, with the longest prediction performed up to 12 time steps. In our work, we attempted to reproduce their method, but the experimental results were not satisfactory. Compared to the Channel Adaptive module, the performance was notably worse, especially for datasets like weather and Etth1. Please refer to Appendix F, specifically Table 11 in the Spatial and Temporal Embedding subsection, for details.
>
>  **W2:**  Why do sampling-based methods easily suffer from the influences of noise? Also, ..., it can still be considered.
>
> **A2:** Here, we want to convey that point-wise sampling methods can be affected if they happen to sample noisy data. MPPN, on the other hand, adopts a multi-resolution smoothing and information extraction approach to capture the underlying multiple periodic patterns in time series. As mentioned in the paper, most time series prediction methods treat time series as sequences similar to speech or natural language, utilizing convolutional or attention-based models. However, these approaches tend to overlook the inherent characteristics of time series, such as periodicity.
>
> As mentioned in [6], attention mechanisms are permutation-invariant. However, time series data is inherently ordered, and removing this sequential order renders the data meaningless. Empirical evidence supports this notion, as MPPN achieves state-of-the-art performance on 9 publicly available time series datasets from diverse domains using a significantly lower number of parameters.
>
> **W3:**  The authors argue that, ... make connections.
>
> **A3:** We think that by smoothing out noise or diminishing its weight, the impact of noise can be reduced. The utilization of the multi-resolution module effectively smooths noise, while periodic sampling helps to extract clean patterns. This approach can be seen as extracting the main patterns while diminishing the influence of noisy patterns.
>
> **W4:**  Why do results in [6] highlight the importance of focusing on the intrinsic properties of time series?
>
> **A4:** [6] have questioned the permutation-invariant self-attention mechanism causes loss of temporal information for LTSF tasks. They have shown that an embarrassingly simple linear model outperforms all Transformer-based models. This highlights the importance of focusing on intrinsic properties of time series.
>
> **W5:**  Can you give the reason why you set kernel and dilation size to  $\frac{L}{Period_i}$ and $\frac{Period_i}{r}$ ?
>
> **A5:** As described in Section 3.3 of the paper, to achieve periodic sampling, we set the kernel size as  $\frac{L}{Period_i}$ and the convolution dilation as $\frac{Period_i}{r}$, where ${Period}_{i}$ represents the length of the period, $L$ denotes the length of the input sequence, and $r$ denotes the chosen time resolution.
>
> **W6:** . In Table 2, the authors provide predictability of each dataset based on Section 3.2. Can you give a more detailed formula for predictability, such as how to identify sub-strings encountered before?
>
> **A6:** The detained formula for predictability is presented in Equation (3) and Equation (4). See Section 3.2 for details. In Equation (3), $\Lambda_i$ denotes the length of the shortest substring starting at position i which does not previously appear from position 1 to i-1. For example, for the array [1,0,0,1,0,0], the corresponding $\Lambda_i$ would be [1,1,2,0,0,0] (the original array starts repeating from the fourth number, so the lengths of the subsequent shortest non-repeating arrays are all zero).

---

> > ### Comment · Reviewer_saoQ · 2023-11-23
> > **Thank you for your effort**
> >
> > Thank you for your effort to resolve my concerns.
> > However, I think huge improvements are required for this paper but the authors' responses and the revised paper are uploaded too late to discuss about many points for improvements. Therefore, I uphold my score as before.

---

### Official Review · Reviewer_fdFp · 2023-10-31

**Soundness:** 2 fair
**Presentation:** 2 fair
**Contribution:** 2 fair
**Rating:** 5
**Confidence:** 4

**Summary:**

This paper proposes Multi-resolution Periodic Pattern Network (MPPN) for long-term series forecasting. It first constructs context-aware multi-resolution semantic units of time series and then employs multi-periodic pattern mining to capture the key patterns of time series. It further proposes a channel adaptive module to capture the multivariate perceptions towards different patterns. In summary, it introduces a straightforward convolutional-based network for time series forecasting, prominently leveraging the multi-scale periodic bias.

**Strengths:**

Generally good and robust empirical studies. It demonstrates that the proposed algorithm achieves comparable performance to recent algorithms in similar approach, such as TimesNet. It also shows slightly better or comparable performance to PatchTST.

**Weaknesses:**

1.	As the proposed method share many similarities to TimesNet, it is highly suggested to provide a clear discussion on the distinction between the proposed algorithm and TimesNet. It is crucial to emphasize the uniqueness and contribution of this work.
2.	It is suggested to review the consistency of the experiment numbers and their corresponding claims. The statement "Compared to the up-to-date Transformer-based models, MPPN achieves substantial reductions of 22.43% in MSE and 17.76% in MAE" is inaccurate. The numbers presented in Table 3 do not support such a claim. PatchTST, being an up-to-date transformer-based model, shows only marginal improvement. This claim is crucial as it shows how it compares to the recent SOTA, and I would request the authors to make it clear and accurate, especially a direct comparison with TimesNet.
3.	Regarding the time complexity analysis in Figure 5, I generally agree with the author's claim about MPPN's training time efficiency, but I still urge the author to double-check their numbers. It is highly improbable that the training time per step for 192 is smaller than that for 96. In most cases, training times increase as the prediction lengths increase. The author should provide a reasonable explanation for this discrepancy.
4.	The exploration of channel-adapted patterns is intriguing. In the model analysis section, I would recommend including plots of pattern numbers 3, 4, and other notable ones discussed in Figure 4 to Figure 3, in addition to the hourly sampling and 3-hourly averaging. The discussion of periodic patterns is straightforward and I believe there is no need for a dedicated section or figure.
5.	Although I acknowledge the potential of this method, the presentation of this paper can be further improved.

**Questions:**

As stated in the weakness.

---

> ### Author Response · Authors · 2023-11-22
> **Response to Reviewer fdFp**
>
> **W1:** As the proposed method share many similarities to TimesNet, it is highly suggested to provide a clear discussion on the distinction between the proposed algorithm and TimesNet. It is crucial to emphasize the uniqueness and contribution of this work.
>
> **A1:** MPPN and TimesNet are fundamentally different in terms of their overall architecture. The only similarity lies in both utilizing Fourier Transform to calculate the specific period of a given time series, which is indeed a common technique.
>
> In terms of the model architecture, significant distinctions can be observed between MPPN and TimesNet. TimesNet transforms one-dimensional temporal series data into a two-dimensional format, subsequently leveraging a computer vision backbone (involving a two-dimensional convolutional network) for feature extraction. MPPN, in contrast, solely employs one-dimensional dilated causal convolutions for feature extraction without incorporating any two-dimensional convolutional structures. MPPN involves the utilization of multi-resolution patching and periodic pattern mining to extract key features from one-dimensional temporal sequences. In addition, TimesNet operates as a pre-trained model, whereas our model does not undergo pre-training.
>
> Regarding model performance, our proposed MPPN demonstrates an average reduction of over 30% in Mean Squared Error (MSE) compared to TimesNet, and the training time is nearly 10% of that required by TimesNet. Please refer to Table 3 and Figure 5 in our paper for detailed comparisons.
>
> Lastly, considering the complexity of the models, as mentioned in the overall response, for the electricity dataset, TimesNet has a parameter count that is over 500 times larger than MPPN. Similarly, for the Traffic dataset, TimesNet's parameter count is over 1000 times larger than MPPN. This implies that MPPN achieves superior performance in long-term time series prediction with significantly fewer parameters compared to TimesNet.
>
> **W2:** It is suggested to review the consistency of the experiment numbers and their corresponding claims. The statement "Compared to the up-to-date Transformer-based models, MPPN achieves substantial reductions of 22.43% in MSE and 17.76% in MAE" is inaccurate. The numbers presented in Table 3 do not support such a claim. PatchTST, being an up-to-date transformer-based model, shows only marginal improvement. This claim is crucial as it shows how it compares to the recent SOTA, and I would request the authors to make it clear and accurate, especially a direct comparison with TimesNet.
>
> **A2:** In section 4.1 Experiment Settings, we meticulously categorized the baseline methods employed in our paper, distinguishing PatchTST and TimesNet as the “pre-trained models”. Consequently, when referring to "up-to-date Transformer-based models" in our work, it specifically pertains to FEDformer. A comprehensive comparison reveals that, in contrast to TimesNet, our proposed MPPN attains noteworthy reductions of 18.14% in MSE and 10.56% in MAE across all datasets.
>
> **W3:** Regarding the time complexity analysis in Figure 5, I generally agree with the author's claim, ... explanation for this discrepancy.
>
> **A3:** In general, the training time per step for a sequence length of 192 should be expected to exceed that for a length of 96. However,  owing to the efficiency of our proposed method, extending the history window length has a minimal impact on the overall training time. In Figure 5, we presented the training time for a single epoch out  of the total 100 epochs, with some inherent randomness. Besides, we meticulously cross-verified the training times for all other epochs, revealing that the majority of the 192-step training times are marginally longer than their 96-step counterparts.
>
> **W4:** The exploration of channel-adapted patterns, ... a dedicated section or figure.
>
> **A4:** Figure 4 illustrates the channel embedding matrix on ETTh1 dataset with eight patterns during one day. The varying hues and numbers in each block represent the sensitivity of various variates to the eight patterns. For each channel, the importance of patterns is dynamic and constantly changing. If depicted as in Figure 3, the visual representation would appear highly complex. The channnel adaptive module is introduced to adaptively model different perceptions of multivariate series towards various temporal patterns, which help identify which patterns in the input are the most prominent, further enhancing the prediction performance.
>
> **W5:** Although I acknowledge the potential of this method, the presentation of this paper can be further improved.
>
> **A5:** We appreciate your suggestion, and we will thoroughly review the paper to ensure proper grammar, clarity, and expression, thereby avoiding any potential grammatical errors or other issues.

---

> > ### Comment · Reviewer_fdFp · 2023-11-23
> >
> > I appreciate the authors' detailed response and their efforts to address the points raised in my initial review. Their clarifications have provided a clearer understanding of the novelty and methodology of the work, which is indeed promising.

---

### Official Review · Reviewer_FFSt · 2023-11-01

**Soundness:** 3 good
**Presentation:** 3 good
**Contribution:** 2 fair
**Rating:** 5
**Confidence:** 3

**Summary:**

This paper introduces a new model MPPN for the long-term time series forecasting task. It contains 3 key designs: context-aware multi-resolution semantic units (multi-resolution patching) which capture patterns at different granularity, a multi-period pattern mining mechanism deployed by dilated convolution with the selected most significant frequencies from the data, and a channel adaptive module to learn adaptive weights on the mined temporal patterns. Besides, the authors evaluate entropy-based predictability for the time series datasets as an upper bound of accuracy. The experiment results show the effectiveness of the model against the other baselines, and the ablation study reveals the importance of each design proposed in the model. Also, the training time of the model is less than most of the competitors, indicating its efficiency in handling long-term time series forecasting task.

**Strengths:**

This paper is generally well-written, and I list some of the strengths here:

1. The motivation is clear. The paper emphasizes its focus on treating time series as an overlay of multi-resolution patterns. By doing so, the proposed Multi-Periodic Pattern Network (MPPN) model aims to capture multi-resolution and multi-periodic patterns in time series data—a strategy that is highly applicable in real-world scenarios. The implementation of this idea is also straightforward and clear.

2. Pre-evaluation of predictability. One of the novel contributions is the introduction of an entropy-based metric designed to measure the predictability of a given dataset. This offers a theoretically grounded upper bound for potential forecasting accuracy, thereby serving as an insightful metric for evaluating data quality.

3. The forecasting results are good. The MPPN model delivers superior performance in multivariate time series forecasting, outperforming state-of-the-art baseline models in both MSE and MAE metrics. This speaks volumes about the model's forecasting capabilities.

4. The paper includes a methodically structured ablation study. It demonstrates that the model with all three components integrated performs optimally, thus validating the significance of each individual module.

5. The training time of the model is better than all the baseline models besides DLinear, indicating a good efficiency of the model. The main reason could be the well-designed framework, as well as the CNN-based framework being potentially more efficient than large Transformer-based models.

6. The paper is well-written and the materials are organized well, offering readers an accessible and comprehensible overview of the proposed model.

**Weaknesses:**

1. Cross-channel/Spatial correlation: The cross-channel dependency is not explicitly discussed in this paper. MPPN mainly focus on mining the temporal patterns, and aggregating them for different variables. Although this indicates some relationship between different channels, it is not clear how the correlation of different channels / patterns is modeled or learned by the model.

2. Multiple resolution selection: In the current formulation, the multiple resolutions are selected as a fixed set, lacking a generalized criterion for selection. While this may be acceptable in cases where periodicity is well-understood, it becomes a limitation for datasets with unclear periodicities (e.g., exchange rate data). I suggest incorporating an adaptive resolution selection mechanism, particularly for cases where no prior knowledge about the dataset is available.

3. Code / Open source: The experiment details discussed in this paper are limited. It would be beneficial for the authors to provide an anonymous link to the code or a more detailed implementation guide to bolster the paper's credibility.

4. Marginal improvement in ablation study: Although the comprehensive model highlighted in Table 4 outperforms the alternatives, the gains attributed to periodic sampling appear to be marginal. I recommend that the authors conduct additional robustness tests on this particular aspect or demonstrate how this module contributes to other performance metrics, such as training efficiency.

5. Period pattern motivation: The general idea of mining multi-resolution and multi-periodic patterns is compelling. However, Figure 3 is used to claim the motivation of mining periodic pattern from the data, which might need more discussion. Specifically, the magnitude of the fluctuations at different time points may not directly indicate a clear periodic pattern. There is a lack of illustration of how this model incorporates the variance issue besides the periodicity of mean.

**Questions:**

1. The novelty of the multi-period and multi-resolution design may need to be discussed. The ideas of multi-periodicity and multi-resolution are not new, so I am interested in the main novelty of this work, especially when compared to previous works that employ similar ideas.

2. On page 7, you claim a 22.43% (MSE) and 17.76% (MAE) reduction in error compared to Transformer-based models. These numbers don't seem to correspond clearly with the data presented in Table 3. Could you clarify the methodology behind these calculations?

3. Your use of an entropy-based metric for measuring predictability is intriguing but raises questions. The DLinear paper (https://arxiv.org/pdf/2205.13504.pdf) suggests that simplistic "Repeat" models can outperform others in exchange rate datasets, which might indicate a certain level of predictability without providing actionable insights. My expectation is that predicting the daily change / return of exchange rate would be much harder since there is no such “Repeat” effect. Given that your paper shows relatively high predictability scores for exchange rates, could you comment on whether the "Repeat" effect may have influenced these scores?

4. The elements in the heat map of Figure 4 don't show significant variations, especially for HUFL where most values hover around 0.5. Could you provide further insights into how to interpret this information, especially when analyzing the model's sensitivity to various temporal patterns?

5. Your MPPN model aims to mine multi-resolution and multi-periodic patterns. In contrast, Transformers use multi-head attention which could potentially capture multiple patterns with different periodicities. Could you offer an analysis that delineates the differences and possibly the advantages of MPPN over multi-head attention?

---

> ### Author Response · Authors · 2023-11-22
> **Response to Reviewer FFSt (Weaknesses)**
>
> **W1**: Cross-channel/Spatial correlation: The cross-channel dependency is not explicitly discussed, ... by the model.
>
> **A1**: To be honest, our MPPN method does not explicitly model cross-channel correlations. We primarily focus on modeling the characteristics of each individual channel through the Channel Adaptive Module. We have experimented with over 20 different approaches to capture inter-channel correlations, but the effectiveness varied across different datasets. Therefore, we believe it is essential to first assess whether there are indeed correlations between channels. Otherwise, applying common techniques such as channel attention for cross-channel correlation modeling might introduce unwanted noise patterns. We found that most channels are primarily influenced by only a few others.
> In our future work, we plan to further explore this direction and investigate the presence and nature of inter-channel correlations in multivariate time series data.
>
> **W2**: Multiple resolution selection: In the current formulation, ... dataset is available.
>
> **A2**: Thank you for your suggestion. In our method, the concept of resolution is analogous to a smoothing window that helps in filtering out noise and extracting general multi-resolution and multi-periodic patterns. Therefore, we have chosen a fixed set of commonly used resolution windows, similar to selecting convolutional kernel sizes in convolutional networks.
> However, we acknowledge that the extracted patterns corresponding to different resolution settings may have distinct objective physical meanings. In the future, we will consider your suggestion and explore the design of an adaptive resolution selection mechanism.
>
> **W3**: Code / Open source: The experiment details, ... the paper's credibility.
>
> **A3**: We have supplemented the source code as requested. Please refer to the overall response for further details.
>
> **W4**: Marginal improvement in ablation study: Although the, ... training efficiency.
>
> **A4**: We appreciate your question, and we would like to provide an explanation. Removing the Periodic Sampling Module would significantly increase the number of model parameters. Additionally, the Periodic Sampling Module contributes a slight improvement in performance. Therefore, we made the decision to include this module in the final architecture. The Periodic Sampling Module ensures that the parameter count of MPPN does not significantly increase with longer input and output sequences, resulting in a lightweight yet effective model.
> We appreciate your suggestion, and we will incorporate additional experimental evidence in the appendix to demonstrate the contribution of this module towards improving training efficiency metrics.
>
> **W5**: Period pattern motivation: The general idea, ... mean.
>
> **A5**: As shown in Section 3.3, we employ the Fast Fourier Transform (FFT) to analyze the periodicity of the time series, with the systematic presentation of results in Table 3 providing robust evidence for the intrinsic periodic nature of the Electricity dataset. Furthermore, Figure 3 visually reinforces this observation, depicting data with the same sampling frequency exhibiting similar trends at corresponding time periods, while highlighting discernible differences in periodic patterns across across different resolutions. This visual representation not only serves as a validation of our conclusions but also provides a clear elucidation of our multi-resolution modeling approach, as emphasized in Section 3.3.

---

> ### Author Response · Authors · 2023-11-22
> **Response to Reviewer FFSt (Questions)**
>
> **Q1**: The novelty of the, ...previous works that employ similar ideas.
>
> **A1**: Firstly, our research goal is to explore time series pattern learning and demonstrate that accurate pattern capture can lead to superior performance even with simple and lightweight models compared to more complex models (with parameters counts up to 10 or even 100 times larger). As you mentioned, previous works have acknowledged the importance of multi-periodicity and multi-resolution, such as TimesNet and MICN. However, they did not combine these two aspects with periodic sampling for extracting specific time series patterns.
>
> Additionally, we compared the parameter count of several models, as mentioned in the overall response. It can be observed that, in contrast to models like Timesnet, PatchTST, and MICN, MPPN maintains the fewest number of parameters while achieving very competitive performance (refer to Section 4.2 of the paper).
>
> **Q2**: On page 7, you claim a 22.43% (MSE) , ... behind these calculations?
>
> **A2**: In section 4.1 Experiment Settings, we meticulously categorized the baseline methods employed in our paper, distinguishing PatchTST and TimesNet as the “pre-trained models”. Consequently, when referring to "up-to-date Transformer-based models" in our work, it specifically pertains to FEDformer.
>
> **Q3**: Your use of an entropy-based, ... these scores?
>
> **A3**: We agree with the reviewer the "Repeat" effect may have influenced the predictability scores. Time series with pronounced periodicity typically exhibit a higher predictability. However, the nine widely used public datasets often exhibit complicated temporal patterns which may render the situation more intricate. The predictability of time series is often influenced by various factors such as the length of the time series, missing values, stationarity, and periodicity. It is an inherent attribute of a time series and is not influenced by specific forecasting methods. As for the exchange rate, results in Table 3 reveal that MPPN, DLinear and MICN could all achieve satisfactory prediction results which validate the high predictability value in Table 2.  In conclusion, Predictability reveals the theoretical potential of how much a time series can be forecasted, but in practice, it often requires a case-by-case analysis based on specific circumstances.
>
> **Q4**: The elements in the heat map, ... temporal patterns?
>
> **A4**: Figure 4 illustrates the channel embedding matrix on ETTh1 dataset with eight patterns during one day. The varying hues and numbers in each block represent the sensitivity of various variates to the eight patterns. For HULF, although most values hover around 0.5, it can be seen that it is most influenced by the third and fouth patterns, which correspond to the midday time of the day. The channnel adaptive module is introduced to adaptively model different perceptions of multivariate series towards various temporal patterns, which help identify which patterns in the input are the most prominent, further enhancing the prediction performance.
>
> **Q5**: Your MPPN model, ... attention?
>
> **A5**: The objective of MPPN is to extract key temporal patterns from time series data, whereas transformer-based methods aim to capture all perceivable sequence patterns. The structure of MPPN is simpler and its goal is more clearly defined (to mine multi-resolution and multi-periodic patterns in time series). On the other hand, the multi-head attention mechanism is a generalized approach for capturing sequence correlations.
>
> As mentioned in the paper, most attention-based time series prediction methods treat time series data as similar to natural language sequences, overlooking the distinctive characteristics of time series, such as periodicity. As discussed in [1], attention mechanisms are permutation-invariant, but time series data is inherently ordered, and losing its sequential order renders it meaningless. **While the multi-head attention mechanism could potentially capture multiple patterns, critical patterns (such as periodic patterns) might be overshadowed by the other patterns.** In contrast, MPPN explicitly captures multi-periodic patterns with a simple model structure, highlighting the importance of extracting key patterns for representing time series data. The experiments further demonstrate that MPPN achieves state-of-the-art performance on 9 publicly available time series datasets from different domains, despite having a significantly lower number of parameters.
>
>  [1] Zeng, Ailing, et al. "Are transformers effective for time series forecasting?." *Proceedings of the AAAI conference on artificial intelligence*. Vol. 37. No. 9. 2023.

---

### Official Review · Reviewer_qZtm · 2023-11-01

**Soundness:** 3 good
**Presentation:** 3 good
**Contribution:** 3 good
**Rating:** 5
**Confidence:** 4

**Summary:**

This article presents a new architecture for time series forecasting. This architecture consists primarily of two parts: one is responsible for extracting multi-resolution spatial patterns and periodic patterns, and the other is responsible for learning dependencies between signal channels to modulate the previously obtained output. Spatial pattern learning is done through a convolutional network, while periodic pattern learning involves using FFT initially and then dilated convolutions on the previous convolutions. The two outputs are concatenated and modulated by the second module. The second module primarily performs channel embedding, allowing separate modulation for each channel. The article concludes with experiments on 9 common datasets in the field and state-of-the-art baselines. An ablation study is also presented, along with qualitative results.

**Strengths:**

* The paper effectively situates the architecture within the state of the art.
* The literature review is substantial and well-detailed.
* The experiments are well-described, comprehensive, and the qualitative and ablation experiments are very useful.

**Weaknesses:**

* I can't seem to grasp the purpose of section 3.2. As the authors state, 'There exists a multitude of seminal works in the domain of predictability,' and the few lines the authors develop on the subject seem very close to the work of [Xu et al. 2019]. What is the value of introducing this method (and it's not clear if the authors consider it a contribution or not), and what are the differences compared to what already exists? The only use of predictability is at the beginning of section 4.2, where predictability is linked to the experimental results. However, as the authors note, 'Although the situation is not always the case, the general rule is that for datasets with higher predictability, carefully constructed predictive models usually tend to exhibit lower prediction metrics,' which leads to an inconclusive result without further explanation.

* Sections 3.3 and 3.4, the core of the paper, are quite short. The description of the architecture is minimal, and it would have been preferable to clarify and expand these sections significantly to assist the reader.

* No code is provided with the article as supplementary material, which is unfortunate because I am quite curious to run the experiments. The results are indeed surprising: the first module is very similar to convolutional architectures from the state of the art, perhaps even simpler. However, in the ablation study, the results without the adaptation module are much better than competing architectures. In the case of the electricity dataset, adaptation seems to bring no improvement, and only the multi-resolution component appears to have a significant impact, with results far superior to convolutional architectures. This phenomenon is quite perplexing to me, and I would have appreciated more insights on this matter.

**Questions:**

* Can you explain why the performances of this architecture without the channel adaptation module  appear to be better than convolutional architectures similar to this model?

---

> ### Author Response · Authors · 2023-11-22
> **Response to Reviewer qZtm**
>
> **W1:** I can't seem to grasp the purpose of section 3.2. As the authors state, 'There exists a multitude of seminal works in the domain of predictability,' and the few lines the authors develop on the subject seem very close to the work of [Xu et al. 2019]. What is the value of introducing this method (and it's not clear if the authors consider it a contribution or not), and what are the differences compared to what already exists? The only use of predictability is at the beginning of section 4.2, where predictability is linked to the experimental results. However, as the authors note, 'Although the situation is not always the case, the general rule is that for datasets with higher predictability, carefully constructed predictive models usually tend to exhibit lower prediction metrics,' which leads to an inconclusive result without further explanation.
>
> **A1:** In our paper, predictability arises as a prerequiste before performing long-term series forecasting. We derive the predictability based on the widely used entropy rate to circumvent predicting time series with low predictability, such as Gaussian white noise. In fact, we have stated in Section 3.2 that "…we firstly evaluate the predictability following the method in Song et al. (2010),…". Nevertheless, we emphasize the evaluation of predictability before developing a specific time series forecasting method. To the best of our knnowledge, we are the first to derive predictability results (Table 2) of the nine widely-used time series datasets using entropy rates, which has not been fully addressed by previous deep learning-based forecasting methods.
>
> **W2:** Sections 3.3 and 3.4, the core of the paper, are quite short. The description of the architecture is minimal, and it would have been preferable to clarify and expand these sections significantly to assist the reader.
>
> **A2:** Due to space limitations, we have abbreviated Sections 3.3 and 3.4 in our current manuscript. We will make appropriate abridgements in other sections of the manuscript, allowing us to allocate more space for a comprehensive expansion of Sections 3.3 and 3.4.
>
> **W3:** No code is provided with the article as supplementary material, which is unfortunate because I am quite curious to run the experiments. The results are indeed surprising: the first module is very similar to convolutional architectures from the state of the art, perhaps even simpler. However, in the ablation study, the results without the adaptation module are much better than competing architectures. In the case of the electricity dataset, adaptation seems to bring no improvement, and only the multi-resolution component appears to have a significant impact, with results far superior to convolutional architectures. This phenomenon is quite perplexing to me, and I would have appreciated more insights on this matter.
>
> **A3:** Firstly, we appreciate the reviewer for your interest in our work. As mentioned in the overall response, we have provided the source code as requested.
> Secondly, we are delighted to explain the concerns regarding the ablation experiments. In the results, it can be observed that the Channel Adaptive Module played a vital role in the weather dataset. As described in Section 4.3 of the paper, the weather dataset comprises different meteorological indicators such as wind speed and air temperature, which exhibit distinct characteristics. Therefore, the channel adaptation is crucial for capturing personalized patterns. On the other hand, for the electricity dataset, different variables represent the electricity consumption of different users. Despite being different variables, they essentially represent electricity consumption data. In this case, the effects of the Channel Adaptive Module become less pronounced, while the Multi-resolution Component plays a more significant role. We believe that the components of MPPN demonstrate distinct effects depending on the characteristics of the specific dataset. The paper provides two dataset examples, and we will supplement additional analysis in the appendix, which will further facilitate understanding the characteristics and commonalities across various time series datasets.
>
> **Q1:** Can you explain why the performances of this architecture without the channel adaptation module appear to be better than convolutional architectures similar to this model?
>
> **A4:** We are glad to provide an explanation. In our ablation experiments, we systematically remove one module at a time to evaluate its impact on the model's performance. When we remove the Channel Adaptive Module from MPPN, we still retain the multi-resolution component and the periodic sampling module. Therefore, the improved performance of MPPN without the Channel Adaptive Module compared to other convolutional architectures can be attributed to the remaining components, namely the multi-resolution patching and the periodic pattern mining .

---

> > ### Comment · Reviewer_qZtm · 2023-11-22
> > **Acknowledgement of authors' response**
> >
> > I thank the authors for the clarifications provided. I have read in detail their responses to my comments and those of the other reviewers. I still believe that the predictability aspect is not well-addressed: either it deserves a much more extensive development and detailed analysis if there is substance, or as it stands, it does not add much. For example, Electricity shows very low predictability and yet achieves better forecasting results than Weather or exchange in the long term!
> > Furthermore, given the date of the authors' response (the last day of the rebuttal), it is impossible for me to examine the code and experiment as I would have liked to. I take note of the authors' willingness to change sections 3.3 and 3.4 and include new results, but I cannot evaluate without having seen the new version. I appreciate the explanations provided regarding my questions about the ablation study. Overall, and after reading the other reviews, I decided to keep my initial score.

---

### Author Response · Authors · 2023-11-22
**Rebuttal is online**

Dear reviewers,

Thanks for your constructive comments and suggestions again. We have responded to all of your concerns with details. Please review our responses to ensure that your concerns have been adequately addressed, and feel free to follow up with any additional questions you may have. Thanks!

We would like to provide a comprehensive response addressing the common concerns raised by the reviewers.

1. Supplementary code: We have addressed the request and included the source code as an attachment (Supplementary Material).
2. Differentiation from TimesNet [1]: Several reviewers questioned the distinctions between our proposed MPPN and TimesNet. We would like to clarify the differences between these two approaches.

MPPN and TimesNet are fundamentally different in terms of their overall architecture. The only similarity lies in both utilizing Fourier Transform to calculate the specific period of a given time series, which is indeed a common technique.

In terms of the model architecture, significant distinctions can be observed between MPPN and TimesNet [1]. TimesNet transforms one-dimensional temporal series data into a two-dimensional format, subsequently leveraging a computer vision backbone (involving a two-dimensional convolutional network) for feature extraction. MPPN, in contrast, solely employs one-dimensional dilated causal convolutions for feature extraction without incorporating any two-dimensional convolutional structures. MPPN involves the utilization of multi-resolution patching and periodic pattern mining to extract key features from one-dimensional temporal sequences. In addition, TimesNet operates as a pre-trained model, whereas our model does not undergo pre-training.

Regarding model performance, our proposed MPPN demonstrates an average reduction of over 30% in Mean Squared Error (MSE) compared to TimesNet, and the training time is nearly 10% of that required by TimesNet. Please refer to Table 3 and Figure 5 in our paper for detailed comparisons.

3. we would like to highlight that our main focus with MPPN is on extracting key patterns from time series, enabling a simple, lightweight, and effective approach for long-term time series forecasting. We attempt to explicitly model the exclusive characteristics of time series, emphasizing the importance of assessing predictability before making predictions for a given time series. Below is a parameter comparison between MPPN and TimesNet [1], PatchTST [2], and MICN [3]:

| Methods  | Parameters  |             |
| -------- | ----------- | ----------- |
|          | Electricity | Traffic     |
| MPPN     | 275,162     | 232,734     |
| PatchTST | 921,184     | 921,184     |
| MICN     | 25,844,129  | 26,952,638  |
| TimesNet | 150,304,769 | 301,692,958 |

In summary, we hope this response addresses the concerns raised by the reviewers and clarifies the differences and advantages of our proposed MPPN method compared to TimesNet.

[1] Wu, Haixu, et al. "Timesnet: Temporal 2d-variation modeling for general time series analysis." ICLR (2023).

[2] Nie, Y., Nguyen, N. H., Sinthong, P., & Kalagnanam, J. (2022). A time series is worth 64 words: Long-term forecasting with transformers. arXiv preprint arXiv:2211.14730.

[3] Wang H, Peng J, Huang F, et al. Micn: Multi-scale local and global context modeling for long-term series forecasting[C]//The Eleventh International Conference on Learning Representations. 2022.



Best regards,

All of the authors